# ADAPTING LLM AGENTS THROUGH COMMUNICATION

## ABSTRACT

Recent advancements in large language models (LLMs) have shown potential for advanced agents. To support online learning for these agents without extensive human supervision, we propose the Learning through Communication (LTC) paradigm, a novel training approach enabling LLM agents to enhance their skills through interactions with their environments and other agents. Through iterative exploration and updating, LTC empowers the agent to assimilate short-term experiences into long-term memory. To optimize agent interactions for task-specific learning, we introduce diverse communication patterns tailored for both single-agent and multi-agent environments. We evaluated LTC on four datasets: ALFWorld (single-agent), HotpotQA (multi-agent collaboration), Chameleon (multi-agent competition), and GSM8k (multi-agent teacher-student). On *ALFWorld*, it exceeds the instruction tuning baseline by 12% in success rate. On *HotpotQA*, LTC surpasses the instruction tuned Llama-7B agent by 5% in EM score, and it outperforms the instruction tuned 9x larger PaLM-62B agent by 0.6%. On *Chameleon*, the winning rate of LTC agent surpasses the baseline by 3.1%. On *GSM8k*, LTC outperforms the CoT-Tuning baseline by 3.6% in accuracy. The results showcase the versatility and efficiency of the LTC approach across diverse domains. We will open-source our code to promote further development of the community.

## 1 INTRODUCTION

Recent advances in large language models (LLMs) have shed light on human-like LLM agents [33, 6, 65]. Previous works have explored how to employ the LLM as the engine of agents, which is complemented by several key components such as planning, tool use, and memory. For planning, Chain of Thought (CoT[66]) prompts the model to think step by step, decomposing complex tasks into smaller and simpler steps. ReAct [72] integrates reasoning and action within LLM, extending the action space to include both task-specific discrete actions (i.e., retrieval) and language generation. In addition, recent research has shown that LLM agents can be further augmented by using *external tools* to solve problems in interactive environments, which involve using retrieved knowledge [47, 72, 57, 18], math tools [45, 72, 27], and code interpreters [12, 59].

However, adapting LLM agents to specific tasks remains challenging. In order to adapt the general pre-trained LLMs to task-specific environments, existing methods focus on In-context learning (ICL [5, 72, 48]) and supervised fine-tuning (SFT [33]). For example, ICL utilizes few-shot prompts as a form of short-term memory [35, 48] to adapt agents' task-specific ability during inference, and SFT typically requires high quality supervised training data for domain adaptation. However, both of these two approaches don't leverage the rich feedbacks (i.e., positive/negative rewards, error message, hint etc) during the interaction with environment and external tools for online learning.

In this paper, we introduce a new training paradigm Learning through Communication (LTC), allowing the LLM agent to do online learning in new environments and tasks with an iterative pipeline. Each iteration of LTC consists of two distinct phases: (1) the exploration phase where the agent interact with the environments and other agents to collect diverse trajectories and feedbacks, and (2) the updating phase that updates the agent model based on these trajectories.

LTC method is a general learning paradigm that can be applied to diverse agent settings with single-agent and multi-agent environments. To collect communication trajectories and reward signals, we designed three structured communication patterns: (1) **Single-agent Monologue** allows a single agent to generate trajectories in a soliloquy style and get interactive feedback from environments;

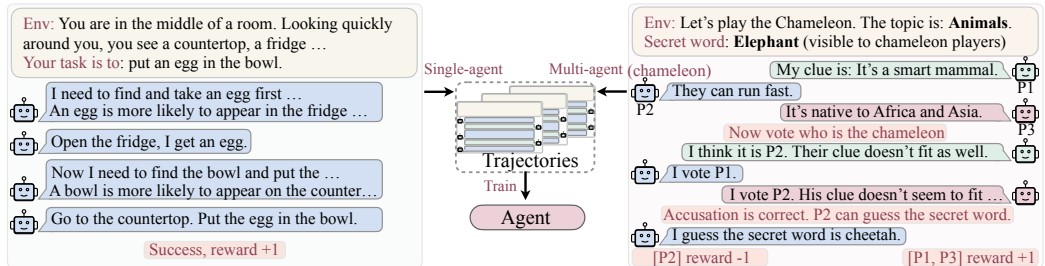

Figure 1: The LTC framework is adept for both single-agent and multi-agent environments. Within these environments, agents have the capability to persistently engage in exploration and interaction to collect trajectories through various communication patterns. Concurrently, LTC facilitates the training of these agents utilizing the data acquired from their exploratory activities. This process enables the agents to autonomously adapt to their respective environments, negating the necessity for human supervision.

(2) **Multi-agent Dialogue** enables the multiple agents interacts with each other and external tools, and utilizes the reward signals provided by the environments; (3) **Teacher-student Dialogue** is a variant of multi-agent dialog that the teacher agent can provide not only linguistic feedback but also the non-linguistic reward signals.

In addition, the updating phase is based on reinforcement learning (RL), and our training loss combines the language modeling loss and the PPO loss [33] to balance the language consistency and reward signals. We also designed a unified replay buffer to store trajectories from diverse environments, where each generated token is treated as an action in the RL formulation. We apply distinct masks to tokens generated by the system, LTC agents, and other agents, respectively. These masks are utilized to apply distinct loss functions to various token types. The replay buffer is updated after each exploration phase and a subset of the buffer is sampled for the updating phase. Through this iterative process, the agent continuously do online learning in new environments.

We evaluate LTC method on three representative datasets: *ALFWorld* for decision-making, *HotpotQA* for knowledge-intensive reasoning, and *GSM8k* for numerical reasoning. Throughout these experiments, LTC consistently outperforms the baselines. In *ALFWorld*, LTC outperforms the strong instruction tuning baseline by 12% on success rate, even in the challenging Pick 2 task. This shows that our communication mechanism enables the agent to learn from its experiences for task solving. On *HotpotQA*, LTC outperforms the instruction tuning baseline by 5% on EM score, and our Llama-7B based agent even obtains slightly better (0.6%) performance than the ReAct-Tuning baseline which uses 9× larger PaLM-62B model. On *GSM8k*, LTC also beats the CoT-Tuning baseline by 3.6% on accuracy. These results highlight the adaptability and effectiveness of LTC approach across varied domains.

Our key contributions are summarized as follows:

1. **Learning through Communication (LTC)**: We propose a new training paradigm named Learning through Communication (LTC), allowing the LLM agent to automatically adapt to new environments and tasks with an iterative pipeline.

2. **Task-specific Communication Patterns**: The LTC paradigm allows flexible design of communication patterns for different tasks. Three specific patterns are introduced: Single-agent Monologue, Multi-agent Dialogue, and Teacher-student Dialogue. These patterns are composable to help generate diverse structured interaction and feedback signals for agent training for different types of tasks.

3. **Empirical Study and Findings**: Through rigorous studies on public benchmark tasks, we demonstrate the effectiveness of LTC, and it could be a superior approach compared with instruction-tuning or prompting baselines.

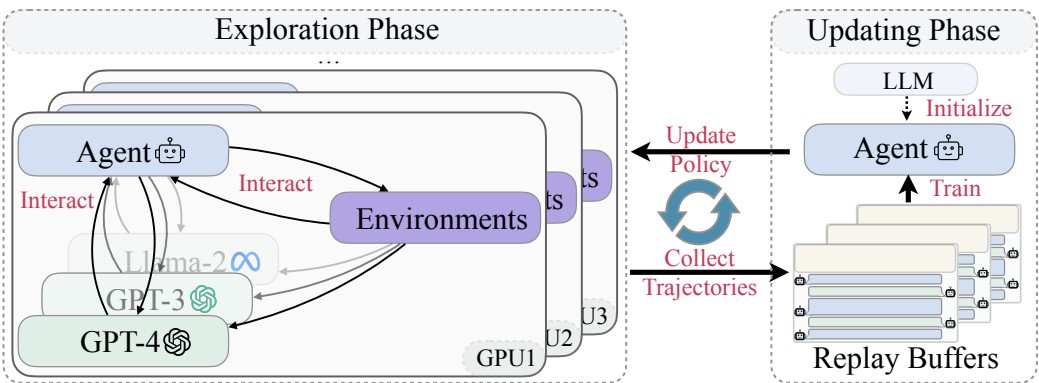

Figure 2: LTC has an iterative two-phase framework. During the exploration phase, the agent proactively explores new environments and communicates with other agents, gathering the trajectories to update the replay buffer. Then the agent is trained for updating the policy in the updating phase.

## 2 RELATED WORK

### 2.1 INSTRUCTION TUNING

Instruction tuning (IT) is an important technique for improving the capabilities and controllability of LLMs [39, 5, 65, 38, 32, 8, 56]. Many studies have been dedicated to instruction data generation and selection [9, 62, 19]. For instance, Unnatural Instructions [14] is created by using the Super-Natural Instructions dataset [63] as a seed to prompt InstructGPT [33]. Self-Instruct [62] employs a recursive pipeline that generates instruction data from hand-crafted seed tasks using ChatGPT [31]. Other studies focus on fine-tuning pre-trained LLMs with instruction data. BLOOMZ [30] is initialized with BLOOM [44] and then fine-tuned using the xP3 instruction dataset [30]. Flan-T5 is initialized with T5 [41] and fine-tuned with the FLAN dataset [25]. Additionally, after the release of LLaMA [56], many works have utilized it as the base model for instruction tuning, such as Alpaca [55], Vicuna [7], and GPT-4-LLM [36]. Some papers explore alignment fine-tuning using RLHF [33, 3, 52, 20]. InstructGPT [33] employs GPT-3 for supervised fine-tuning on a human-filtered instruction dataset, followed by training a reward model and using PPO [46] for RLHF. Claude investigates RLHF [3] and constitutional approaches [4] for making LLMs both harmless and helpful. DPO [40] fine-tunes the LLMs to align with human preferences by directly optimizing a classification problem on preference data instead of RLHF. While these prominent research works focus on aligning LLMs for general instruction-following, our objective is to adapt LLM agents for specific tasks or environments.

### 2.2 LLM AGENTS

LLMs have demonstrated the potential to act as advanced agents [33, 6, 65], and significant progress has been made in developing versatile LLM agents [67, 53, 34, 23, 22, 70] and benchmarks [58, 11, 24]. For planning, Chain-of-Thought (CoT[66]) prompts the model to think step by step, by decomposing complex tasks into smaller and simpler steps. Self Consistency [60, 61] extends CoT by using ensembles of predictions to improve consistency of the LLM. Inner Monologue [17] leverages environment feedback to enhance LLMs' planning and processing capabilities in embodied robotics tasks without extra training. ReAct [72] integrates reasoning and action taking, expanding the action space to include both task-specific discrete actions and language. Reflexion [48] equips agents with dynamic memory and self-reflection capabilities to improve reasoning by using continuous trials in the same environment as feedback. Recent research has also shown that LLMs can be augmented as an autonomous agent to use *external tools* to solve problems in interactive environments. These techniques include retrieval augmentation [47, 72, 18], math tools [45, 72, 27], and code interpreters [12, 59]. Prior works also have explored using multiple LLMs in a collaborative setting to solve complex tasks [13, 37, 21, 64, 54, 1]. Open-source projects like AutoGPT [51], GPT-Engineer [2], and BabyAGI [73] also showcase the potential of LLM not just in generating content but also as a general problem solver. Most of the above methods are based on either human-designed few-shot prompting examples, or finetuning with pre-collected instruction datasets. Our LTC is not a few-shot prompting method and we focus on adapting the agent by collecting training data automatically by exploration.

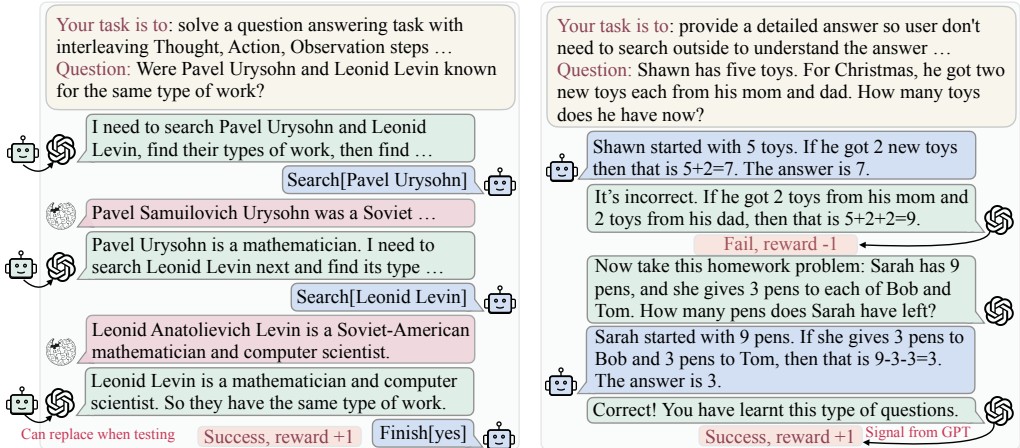

Figure 3: The toy examples to demonstrate communication patterns: 1) the left figure is the Multi-agent Dialogue pattern, where two agent play different roles to collaborate on the task. The thinker agent is responsible for analyzing the situation and give suggestion to the actor agent who is responsible for making decisions. We can just assign the LTC agent to play the thinker agent when testing without GPT-4 agent. 2) the right figure is the Teacher-student Dialogue pattern, where the student agent starts with an initial answer to the current question, and then the teacher directly corrects the answer with a reward. To help the student improve ability instead of just memorizing the solution, the teacher will generate another analogous question to ask the student. Eventually, the student gives a new answer for this analogous question and gets a new reward signal from the teacher.

## 3 LEARNING THROUGH COMMUNICATION

We design Learning Through Communication (LTC), an iterative training method for LLM agents to continuously adapt to new environments. As shown in Figure 2, LTC iterates between two phases: (1) An exploration phase where agents can interact with new environments and other agents to collect trial data with feedback, and (2) a updating phrase to fine-tune the agent to update the policy.

### 3.1 EXPLORATION PHASE

At the start of each iteration, the agent explores the environments to get the trajectories and the reward signal data. We denote these data as a tuple: $\mathcal{S} = (\mathcal{T}, \mathcal{M}, \mathcal{R})$, where $\mathcal{T} = \{t_1, t_2, \ldots, t_n\}$ represents the text data generated by the communication process during agent exploration, $\mathcal{M} = \{m_1, m_2, \ldots, m_n\}$ with $m_i \in \{0, 1, 2\}$ indicates the source of the text data (system or agents), $\mathcal{R} = \{r_1, r_2, \ldots, r_n\}$ with $r_i \in \{-1, 0, 1\}$ represents the reward signals provided by either the system or the agents. We demonstrate the the details of this data structure in Figure 4, $\mathcal{M}$ is the mask list, and $\mathcal{R}$ is the reward list. In PPO training, both the value list and the log-prob list correspond directly to the action list. For brevity, we denote these three lists together as $\mathcal{T}$ here. Please see Appendix A.2 for more details.

To collect the trajectories data $\mathcal{S} = (\mathcal{T}, \mathcal{M}, \mathcal{R})$ from different types of tasks, we design the communication patterns for these tasks. Here we provide three communication patterns:

- **Single-agent Monologue**: Single-agent Monologue is a single-agent soliloquy style communication pattern, designed for general instruction following tasks (Algorithm 1). It split the tasks into step by step like ReAct and CoT, and their own trajectories with system rewards are collected to train themselves at the same time with their exploration. Figure 1 left is a toy example of *ALFWorld* to demonstrate the Monologue pattern with a single agent. This agent soliloquizes to think the situation and take the actions to explore the environment and finally get the reward provided by the environment. This pattern is based on the think and act steps in the ReAct formulation [72], we design the training buffer collection process to make it aligh with our reinforcement learning formulation.

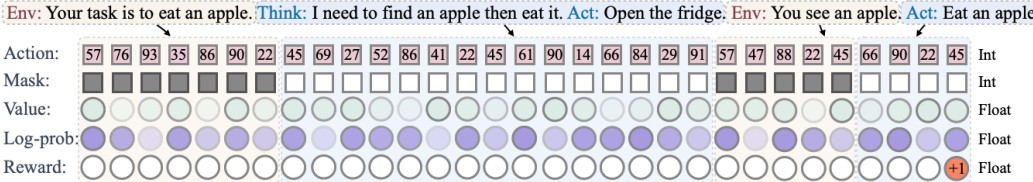

Figure 4: The buffer data is a serial of integer/float sequences. They are collected by the agent's exploration and used for the reinforcement learning phase.

- **Multi-agent Dialogue**: Multi-agent Dialogue is a multi-agent discussion style pattern (Algorithm 2). It is designed for multi-agent collaborating and competing tasks, where multiple agents will play their role by speaking or taking actions in a certain order and **a final reward will be given by the environment** based on the performance of the agents. The left figure of Figure 3 is a toy example of *HotpotQA* to illustrate this pattern for collaborating, where the GPT-4 agent play as a thinker to analyze the situation and give suggestions to the actor agent who is responsible for making decisions. The reward in *HotpotQA* is the correctness of the answer obtained by two agents. And we can use their communication data to train the LTC agents do both thinker and actor so that they can learn how to cooperate with each other to solve the task. The right figure of Figure 1 is a toy example of Multi-agent Dialogue for a competing game task Chameleon, where three agents play different roles. The reward is the win or loss of the game, so they need do with deduction and bluffing in the communication process to win the game. And their games trajectories will be use in LTC iterations to boost the agents.

- **Teacher-student Dialogue**: Teacher-student Dialogue is a teacher-student style pattern for powerful agents to teach the novice agents (Algorithm 3). We design this pattern for complex analytical tasks such as numerical reasoning, which require extensive analytical examples for agents to improve the specific reasoning ability lacking in the pretrained models. Teacher-student Dialogue pattern has two roles (student and teacher) played by two agents, however, in addition to the linguistic feedback, the teacher roles can **directly provide the non-linguistic reward signals**, which are all provided by the system (environments) in the previous pattern. The right figure of Figure 3 is a toy example with GSM8k to demonstrate how the student agent communicates with the teacher agent in a homework-correcting style. In the math question environment, the student agent starts with an initial answer to the current question, then the teacher directly corrects the answer with a reward. To help the student improve ability instead of just memorizing the solution, the teacher will generate another individual question and provide a new reward to the student.

## 3.2 UPDATING PHASE

In the updating phase, the LLM agent model could be optimized through the conversation sessions collected in the exploration stage. Given a example session $\mathcal{S} = (\mathcal{T}, \mathcal{M}, \mathcal{R})$, we mainly utilize two training objects for model training.

- Language model Objective: $\mathcal{L}_{\text{LM}}$ encourages the model to learn from the trajectory $\mathcal{T}$, serving as an unsupervised learning schema to help model for behavior cloning from other agents' response or predicting system feedbacks.
- Reinforcement Objective: $\mathcal{L}_{\text{reinforce}}$ optimizes the model by maximizing the expectation reward provided by environment or a teacher agent (i.e., GPT-4 [32]). It is an goal-oriented objective, and allows the model to learn through both positive and negative signals in the communication session.

Thus, the overall training objective for LTC combines the above two terms:

$$\mathcal{L}_{\text{LTC}}(\mathcal{S}) = \beta \mathcal{L}_{\text{LM}}(\mathcal{T}) + \mathcal{L}_{\text{reinforce}}(\mathcal{S}), \tag{1}$$

where $\beta$ is a balancing hyper-parameter. The off-policy PPO algorithm [46] is utilized for optimizing $\mathcal{L}_{\text{reinforce}}(\mathcal{S})$, and it can be further breakdown into policy loss, value loss and policy entropy regular-

ization terms in implementation. The vanilla PPO algorithm takes the triplet $(state, action, rewards)$ for training. In this case, we sample from the trajectories $(\mathcal{T}_{<i}, t_i)$ for simulating the state-action pairs, specifically, we only keep the tokens generated by agent model itself as actions for policy updating.

# 4 EXPERIMENTS

## 4.1 DATASETS

We conducted experiments on four datasets: *ALFWorld* [50], *HotpotQA* [71], Chameleon [69] and *GSM8k* [10]. Each of these datasets represents a different environment type, namely single-agent, multi-agent collaborating, multi-agent competing, and teacher-student, respectively. And different communication patterns are used: Single-agent Monologue for *ALFWorld*, Multi-agent Dialogue for *HotpotQA* and Chameleon [69], and Teacher-student Dialogue for *GSM8k*.

***ALFWorld***    *ALFWorld* (Figure 1) is a text-based game that follows the ALFRED benchmark [49]. In this game, agents are presented with six types of tasks that involve navigating a simulated household environment using textual actions. With over 50 locations to explore, these tasks demand strategic planning and thorough exploration. Following [50], we utilize the train set that consists of 3553 environments for training our model and the baselines; and we use the unseen test set that comprises 134 environments for evaluatation.

***HotpotQA***    *HotpotQA* is a question-answering dataset that focuses on multi-hop reasoning based supporting facts, with the goal of improving the explainability of QA systems. In this dataset, agents are required to reason across two or more Wikipedia passages to derive answers. We initialize the environments using only the text of the questions, meaning that agents are provided with the question and task description but do not have access to supporting paragraphs. To support their reasoning, agents must either rely on their internal knowledge or interact with an external Wikipedia tool to retrieve the necessary information. For training, we sample the environments from the training set, which consists of 90,447 QA-pairs. For evaluation, we run 500 random examples from the test set, following [72].

***Chameleon***    Chameleon is a multi-player social deduction game environment implemented by the ChatArena [69]. There are two roles in the game, chameleon and non-chameleon. The topic of the secret word will be first revealed to all the players. Then the secret word will be revealed to non-chameleons. Non-chameleons try to identify the chameleon without giving away the secret word, while the chameleon tries to blend in and guess the word. The game involves giving clues, voting on who the chameleon might be, and a final guess from the accused chameleon. We use [3, 4, 5] players setting to train and test the agents' performance.

***GSM8k***    The *GSM8k* dataset is a collection of 8.5K math problems for grade school students. These problems have been crafted by human experts to ensure linguistic diversity. The dataset is divided into two sets: 7.5K problems for training and 1K problems for testing. Each problem in the dataset requires 2 to 8 steps of reasoning to arrive at the solution. The problems primarily focus on fundamental arithmetic operations like addition, subtraction, multiplication, and division.

## 4.2 SETTINGS

**Model Architecture**    We use a modified version of Llama [56] as the base model. To generate state values corresponding to the action tokens, we introduce an additional linear layer to serve ast the value head. This value head acts as an auxiliary output module, and the output values are processed using the $tanh()$ function to ensure they fall within the range of (-1, 1). This adaptation for RL has also been discussed in prior studies [43].

**Agent Pre-training**    We use the Llama-7B model [56] for our LLM agent. To enhance the agent's ability to follow task-specific instructions, we initialize it by instruction fine-tuning (IT). And this initialized agent works as the baseline for a fair comparison. This step is crucial because the original Llama-7B model, without prior instruction fine-tuning, struggled to follow task instructions and generation sensible actions in the environments. To collect data for instruction fine-tuning, we employ GPT3/4 as our agent to explore the environments created from the training set. We then filter out

negative examples and retain positive examples to train the initial agent. For both the *ALFWorld* and *HotpotQA* datasets, we leverage GPT3 (specifically, text-davinci-003). However, for the *GSM8k* dataset, we use GPT4 due to GPT3's inadequate performance in handling mathematical problems, which resulted in a scarcity of positive examples.

**Training details**   We utilize the AdamW optimizer [26] with a batch size of 32. The learning rate is set to 2e-4. In each iteration, the sizes of new environments for agents to explore are: 256 for *ALFWorld*, 512 for *GSM8k*, and 1024 for *HotpotQA*. For parameter-efficient fine-tuning, we employ LoRA [15] with hyperparameters $R = 16$ and $\alpha = 16$. For distributed training, we utilize 4 nodes with 8×A100 GPUs on *HotpotQA* and *GSM8k*. For the experiments on *ALFWorld*, we use 1 node with 2×A100 GPUs due to the dataset's small scale.

**Baselines**   We compare the agents trained by LTC with existing prompting and instruction tuning methods, including ReAct [72], ReAct-IM [17], CoT [66], CoT-SC [60, 61], BUTLER [28]. The detailed of these baselines are described in Appendix A.7. Most of these methods focus on few-shot prompting, and different pre-trained models are used. To ensure a fair comparison, we include the additional baselines named ReAct-Tuning and CoT-Tuning by fine-tuning the Llama-7B model using the collected trajectories as fine-tuning data. In addition, GPT-4 are not used in the test time, and all the results reported are obtained by the trained agent itself.

## 4.3   RESULTS

*ALFWorld*   As shown in Table 1, LTC outperforms the previous best methods* on all of tasks of *ALFWorld*. We can see that Instruction Fine-tuning is already a strong baseline outperforming others, yet our LTC achieves a success rate of 91%, remarkably outperforming the best Instruction Tuning baseline (78%). Notably, on both Cool and Look tasks, LTC obtains a 100% success rate. Even on the hardest Pick Two & Place task (e.g., "put two pencils in the drawer"), it achieves a de-

| Method \ Task | Pick | Clean | Heat | Cool | Look | Pick 2 | All |
|---|---|---|---|---|---|---|---|
| ReAct (avg) | 65 | 39 | 83 | 76 | 55 | 24 | 57 |
| ReAct (best of 6) | **92** | 58 | **96** | 86 | 78 | 41 | 71 |
| ReAct-IM (avg) | 55 | 59 | 60 | 55 | 23 | 24 | 48 |
| ReAct-IM (best of 6) | 62 | 68 | 87 | 57 | 39 | 33 | 53 |
| BUTLER$_g$ (best of 8) | 33 | 26 | 70 | 76 | 17 | 12 | 22 |
| BUTLER (best of 8) | 46 | 39 | 74 | **100** | 22 | 24 | 37 |
| ReAct-Tuning (avg) | 83 | 91 | 91 | 90 | 72 | 8 | 77 |
| ReAct-Tuning (best of 3) | **92** | **97** | **96** | 95 | 78 | 24 | 78 |
| LTC (avg) | 89 | 91 | 93 | 97 | 96 | 67 | 90 |
| LTC (best of 3) | **92** | **97** | **96** | **100** | **100** | **76** | **91** |

Table 1: AlfWorld success rates (%) for 6 tasks. The results of the bottom block are obtained by fine-tuning Llama-7B model.

cent 76% success rate. The Pick Two task requires the agent to perform two sequences of "pick and place" actions in one task, while keeping track of the desired type and the location. The combined sequences and the need to remember the previous location make this task challenging. This may be the reason why baselines achieve lower success rates on this task. In contrast, our LTC agent, which further trains the agent with self-exploration significantly outperforms other agents. This underscores the effectiveness of the communication mechanism in LTC.

*HotpotQA*   As shown in Table 2, LTC outperforms the instruction tuning baseline[†] by 5% on Exact Match (EM) score, and it even outperforms ReAct and CoT on their default settings. Note that ReAct and CoT use PaLM-540B and GPT3-175B as the pre-trained LM model, which is 77x and 25x larger than our the Llama-7B model we used. By sampling 21 CoT trajectories during inference and adopting the majority answer, CoT-SC is slightly better (0.2%) than LTC, and their combined method ReAct → CoT-SC surpasses LTC by 1.9%. Compared to other models with tuning, our Llama-7B based agent even obtains slightly better (0.6%) performance than the ReAct-Tuning baseline with 9× larger PaLM-62B model.

---

*For *ALFWorld*, ReAct and ReAct-IM results are from Table 3 of [72]. BUTLER and BUTLER$_g$ results are from Table 4 of [50], and they are trained with DAgger [42].

[†]For HotPotQA, Prompting method results without fine-tuning are from Table 1&5 of [72]. PaLM-8B and PaLM-62B scores are estimates from Figure 3 of [72].

| Model | Method | EM score |
|---|---|---|
| PaLM-540B | CoT [66] | 29.4 |
| | CoT-SC [60] | 33.4 |
| | ReAct [72] | 27.4 |
| | ReAct → CoT-SC | 35.1 |
| GPT3-175B | ReAct | 30.8 |
| PaLM-62B | ReAct-Tuning | 32.6 |
| | CoT-Tuning | 25.2 |
| PaLM-8B | ReAct-Tuning | 25.0 |
| | CoT-Tuning | 14.0 |
| Llama-7B | ReAct-Tuning | 28.2 |
| | LTC(single-agent monologue) | 31.0 |
| | LTC(multi-agent dialogue) | **33.2** |
| Llama2-13B | ReAct-Tuning | 33.8 |
| | LTC(multi-agent dialogue) | **35.8** |

Table 2: EM scores on HotpotQA with prompt and tuning methods. Methods that use fine-tuning are marked by "-Tuning".

| Model | Method | Accuracy |
|---|---|---|
| PaLM-540B | CoT [66] | 56.5 |
| | CoT-SC [60] | 74.4 |
| GPT3-175B | CoT [66] | 60.1 |
| | CoT-SC [60] | 78.0 |
| Llama-7B | CoT [56] | 11.0 |
| | CoT-SC [56] | 18.1 |
| Llama-7B | CoT-Tuning | 37.7 |
| | LTC(single-agent monologue) | 39.6 |
| | LTC(teacher-student dialogue) | **41.3** |

Table 3: Accuracy on GSM8k. The results of the bottom block are obtained by fine-tuning LLaMA-7B model, while the others are prompting methods without fine-tuning.

**Chameleon** As shown in Table 4, LTC outperforms the instruction tuning baselines by 3.1% on winning rate against GPT-4 players. In the training, all the players are played by the same Llama2-7B model that we are training. While in the testing, to get the winning rate of our trained agent against GPT4, only 1 player is randomly picked to use our trained agent as backend, and other players are played by GPT4. We could see that the LTC agents winning rate improves with the increasing of number of players, we explain this by the more players, the higher chance that the GPT4 players carry the game.

| Method \ #players | n=3 | n=4 | n=5 | overall |
|---|---|---|---|---|
| Llama-Tuning | 20.8 | 20.3 | 23.8 | 21.9 |
| Llama-LTC | **22.9** | **23.4** | **27.5** | **25.0** |

Table 4: Chameleon game winning rates (%) of different number of players settings. At each game, one player is played by target evaluated model, and the others are played GPT-4.

**GSM8k** As shown in Table 3, LTC(teacher-student dialogue) outperforms the instruction fine-tuning baseline by 3.6% on accuracy, and it surpasses the LTC(single-agent monologue) baseline, which does not use the reward and feedback from GPT-4. However, LTC underperforms CoT and CoT-SC with the much larger models (PaLM-540B and GPT3-175B). This phenomenon is because numerical reasoning requires a larger model size and sufficient pretraining data, as observed in [32]. Unfortunately, due to computational resource limitations, we can only train the relatively small Llama-7B model but were unable to train larger-scale models. Nevertheless, we believe that exploring LTC with larger models is promising for future research.

## 5 DISCUSSION

**Efficiency** As mentioned above, prompting-based methods such as ReAct [72] and CoT [66] use a subset of exemplary trajectories from the given task as few-shot prompts during inference. However, these few-shot prompts are often long, which leads to increased inference cost and limited context length for user queries. As shown in Table 5, we compare the number of input tokens for each task. We compute the CoT prompts for **GSM8k**, and we use ReAct for the other two tasks. All the few-shot prompts are sourced from the original paper. As shown, our LTC agents used

| Method | GSM8k (CoT) | Hotpot-QA (ReAct) | Alfworld (ReAct) |
|---|---|---|---|
| ICL | 836 | 1937 | 1744 |
| LTC | **107** | **167** | **189** |

Table 5: Average number of tokens of the input prompts on test sets. LTC does not use any few shot examples in the prompt, hence uses only a fraction of tokens compared to ICL.

only 12.8%, 8.6%, and 10.8% of the input tokens required by the ICL methods on the three tasks, respectively.

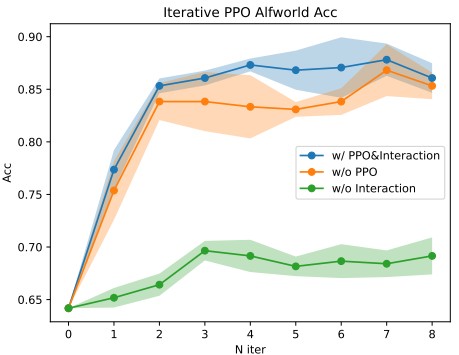 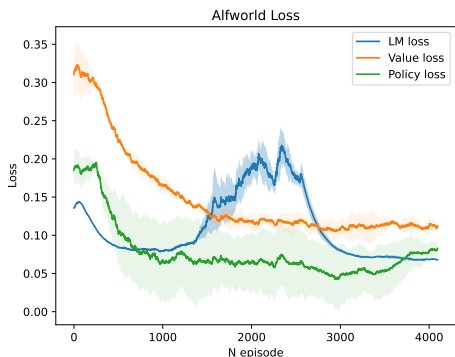

Figure 5: The accuracy curves of PPO training.  Figure 6: The loss curves of PPO training.

**Shortcuts**   One interesting observation is that the GPT-4 agent sometimes employs "shortcuts" to solve problems when serving as a teacher to generate new training data. These shortcuts rely on the internal knowledge acquired during its pretraining process. To illustrate this, we present a case study from *HotpotQA* in Figure 7. In this case, the GPT-4 agent quickly retrieves the answer by leveraging its memorized knowledge about the second entry after receiving the Wikipedia page of the first entry. On the other hand, the bottom of Figure 7 demonstrates a comparison with LLaMA-7B, which was trained using our LTC method with the GPT-4 agent in the loop. LLaMA-7B does not employ shortcuts and instead performs a search for the second entry. This case study demonstrates that communication mechanism in LTC provide additional benefits during learning, compared to soley relying on data generated by GPT-4.

**Ablation**   We conducted ablation studies on the loss design of LTC. Figure 5 illustrates the success rate of agents on the *ALFWorld* dataset under different loss settings. Without using our communication pattern for interactions and merely sampling pre-collected instruction data for training, the improvement was limited. However, when we incorporated our communication pattern to gather data, the model's performance quickly surpassed 80%. Furthermore, employing PPO loss to handle positive and negative samples separately resulted in faster and more significant improvement (blue line). In Figure 6, we present the separate curves of the three main losses during training. Initially, the LM loss showed a decreasing trend. Interestingly, as training iterations progressed, both the value loss and policy loss gradually decreased, which possibly causes the LM loss to increase temporarily. After the value loss and policy loss reached a certain threshold, the LM loss continued to decrease till convergence.

## 6   CONCLUSION

We introduced Learning-Through-Communication (LTC), a paradigm that adapts LLM agents to new tasks and environments via communication-based iterative learning. Within this LTC framework, we have designed three communication modes for common tasks including decision-making, knowledge-intensive reasoning, and numeric reasoning. These communication modes facilitate interactions between LLM agents and their environments, as well as other agents such as GPT-4 and humans. The history of these interactions can be autonomously organized into training data for PPO training so that the agent can adapt to the new task. Our approach represents a closed loop where the agent self-interacts with the environment or other agents, and learning to improve itself with minimal human intervention. Empirically, we have demonstrated that LTC performs strongly in success rate and efficiency across four different tasks: AlfWorld, HotpotQA, Chameleon, and GSM8k. It consistently outperforms existing LLM agent and instruction tuning baselines, showing the promise of the LTC paradigm in adapting LLM agents to new tasks and environments with minimal human effort. As for future work, we plan to explore more diverse communication patterns for different tasks, and involve the communication with human during the iterative learning process. We will open source our code to facilitate further research in this line.

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

**Algorithm 1** The Python-style algorithm to demonstrate Monologue pattern

```python
# agent: LLaMA agent
# input: Task description
# output: S = (T, M, R)

# initialization
T, M, R = [input], [0], [0]

i = 0
while i < max_steps:
    T += ["think:"]
    thought = agent.api(T)
    T.append(thought)
    M.append(1)  # agent message mask
    R.append(0)

    T += ["act:"]
    action = agent.api(T)
    T.append(action)
    M.append(1)  # agent message mask
    R.append(0)

    response = env.excute(action)
    reward = parse(response)
    T.append(response)
    M.append(0)  # system message mask
    R.append(reward)

    i += 1
    if reward != 0:
        break
S = (T, M, R)
return S
```

[66] Jason Wei, Xuezhi Wang, Dale Schuurmans, Maarten Bosma, Ed Chi, Quoc Le, and Denny Zhou. Chain of thought prompting elicits reasoning in large language models. *arXiv preprint arXiv:2201.11903*, 2022.

[67] Lilian Weng. Llm-powered autonomous agents. *lilianweng.github.io*, Jun 2023. URL https://lilianweng.github.io/posts/2023-06-23-agent/.

[68] Ronald J Williams. Simple statistical gradient-following algorithms for connectionist reinforcement learning. *Machine learning*, 8:229–256, 1992.

[69] Yuxiang Wu, Zhengyao Jiang, Akbir Khan, Yao Fu, Laura Ruis, Edward Grefenstette, and Tim Rocktäschel. Chatarena: Multi-agent language game environments for large language models. https://github.com/chatarena/chatarena, 2023.

[70] Binfeng Xu, Zhiyuan Peng, Bowen Lei, Subhabrata Mukherjee, Yuchen Liu, and Dongkuan Xu. Rewoo: Decoupling reasoning from observations for efficient augmented language models. *arXiv preprint arXiv:2305.18323*, 2023.

[71] Zhilin Yang, Peng Qi, Saizheng Zhang, Yoshua Bengio, William W. Cohen, Ruslan Salakhutdinov, and Christopher D. Manning. HotpotQA: A dataset for diverse, explainable multi-hop question answering. In *Conference on Empirical Methods in Natural Language Processing (EMNLP)*, 2018.

[72] Shunyu Yao, Jeffrey Zhao, Dian Yu, Nan Du, Izhak Shafran, Karthik R Narasimhan, and Yuan Cao. React: Synergizing reasoning and acting in language models. In *The Eleventh International Conference on Learning Representations*, 2023. URL https://openreview.net/forum?id=WE_vluYUL-X.

[73] yoheinakajima. Babyagi. https://github.com/yoheinakajima/babyagi, 2023. GitHub repository.

# A APPENDIX

## A.1 COMMUNICATION PATTERNS

To collect the trajectories and the reward signal data from different types of tasks, we design the communication patterns for these tasks and unified the data format as described in Figure 4. Here we use three python-sytle algorithms (Algorithm 1 Algorithm 2 Algorithm 3) to demonstrate how three types of communication patterns help the agent collect exploration data.

---

**Algorithm 2** The Python-style algorithm to demonstrate Dialogue pattern

---

```python
# agent1: LLaMA agent
# agent2: GPT-4 agent
# input: Task description
# output: S = (T, M, R)

# initialization
T, M, R = [input], [0], [0]

i = 0
while i < max_steps:
    T += ["think:"]
    thought = agent2.api(T)
    T.append(thought)
    M.append(2) # teacher agent message mask
    R.append(0)

    T += ["act:"]
    action = agent1.api(T)
    T.append(action)
    M.append(1) # student agent message mask
    R.append(0)

    response = env.excute(action)
    reward = parse(response)
    T.append(response)
    M.append(0) # system message mask
    R.append(reward)

    i += 1
    if reward != 0:
        break
S = (T, M, R)
return S
```

---

## A.2 BUFFER STRUCTURE

The communication data will be saved as replay buffers for the updating phase, and the buffer data format is a serial of tokens sequences demonstrated in Figure 4. We treat each token as the action unit in our reinforcement learning formula, and each exploration trail is processed into 5 data sequences $[\mathbf{S_a}, \mathbf{S_m}, \mathbf{S_v}, \mathbf{S_l}, \mathbf{S_r}]$:

- $\mathbf{S_a}$: A list of integers representing the generated token ids encoded by the tokenizer. All the valid text trajectories are recorded as a queue, including system texts like environment descriptions, feedback, and agent texts like parsed actions, thinking processes, and hints from other agents. While the invalid generated text of the agent will be skipped, such as nonsense string and action text can not be parsed. These tokens are treated equally as the input for the LLM, but they have different masks to apply different losses.

- $\mathbf{S_m}$: The system mask to mask different types of input tokens to control the training loss. We set 0 as the default mask for system texts like environment descriptions, system feedback, and system prompts, the actions encoded from these kinds of texts are not actions we want the agent to learn, so they will be masked out both policy loss and value loss in the PPO algorithm. We set 1 as the mask for agents-generated tokens like the keywords of decisions and the thinking process, which are the main supervising objects of our reinforcement learning pipeline, so they will be assigned full policy loss and value loss. We set 2 as the mask for hints or feedback from other agents, which are the actions we also want our own agent to learn but without instant state values since they are not generated by our agent. So the tokens with mask 2 will be mask out only the value loss and supervised by the policy loss.

**Algorithm 3** The Python-style algorithm to demonstrate Analogue pattern

```
# agent1: LLaMA agent
# agent2: GPT-4 agent
# input: Question description
# output: S = (T, M, R)

# initialization
T, M, R = [input], [0], [0]

i = 0
while i < max_steps:
    T += ["answer_the_question_step_by_step:"]
    answer1 = agent1.api(T)
    query = T + answer1 + ["the_answer_is_correct,_yes_or_no?_also_gives_a_better_answer"]
    response = agent2.api(query)
    reward, answer2 = parse(response)
    T.append(answer1)
    T.append(answer2)
    M.append(1) # student agent message mask
    M.append(2) # teacher agent message mask
    R.append(reward)
    R.append(+1) # assume teacher is correct

    query = query + response + ["please_generate_a_similar_qa_pair_to_teach_the_student:"]
    response = agent2.api(query)
    new_question, teacher_answer = parse(response)
    new_question += "answer_the_question_step_by_step:"
    student_answer = agent1.api(new_question)
    reward = parse(student_answer, teacher_answer)
    T.append(new_question + student_answer)
    M.append(1) # student agent message mask
    R.append(reward)

    i += 1

S = (T, M, R)
return S
```

- **$S_v$**: The state values corresponding to the actions obtained by the value head our the agent model. The value head is an addition layer inserted to the original pre-trained LLM architecture, we implement it by inserting a linear layer after the second-to-last LlamaDecoderLayer as the auxiliary output module and the output values are processed by a $tanh()$ function to keep it range inside $(-1, 1)$.

- **$S_r$**: The rewards corresponding to the actions. The rewards are very sparse, most of the actions are zero-reward, and only when the current task is finished or the token length of the current buffer has just overflowed it will be non-zero value: +1 for positive, -1 for negative.

## A.3  ALGORITHM OF LTC

The implementation of LTC can be summarized as Algorithm 4, we unveil the structural framework that embodies the Learning Through Communication (LTC) paradigm, meticulously crafted to facilitate autonomous and progressive learning through iterative cycles. At the outset, the pre-trained Large Language Model (LLM) agent undergoes a fine-tuning phase to align its initial configuration with the intended learning tasks. Following this, an exploration phase is initiated where a batch of training environments is sampled and subsequently engaged by the agent to generate trial data asynchronously, leveraging the power of parallel computing across multiple GPUs to enhance efficiency. This newly generated data is synchronized across all GPUs to foster a cohesive learning base, which is stored in a replay buffer for further utilization. As the agent iterates through this process, it continually samples from this replay buffer during the updating phase, employing the Proximal Policy Optimization (PPO) algorithm in a distributed data parallel (DDP) setting to refine its strategies and adapt dynamically. This code snippet, therefore, encapsulates the innovative asynchronous and distributed nature of the LTC paradigm, marking a significant stride in fostering intelligent, adaptive, and collaborative artificial intelligence agents.

## A.4  TRAINING LOSS

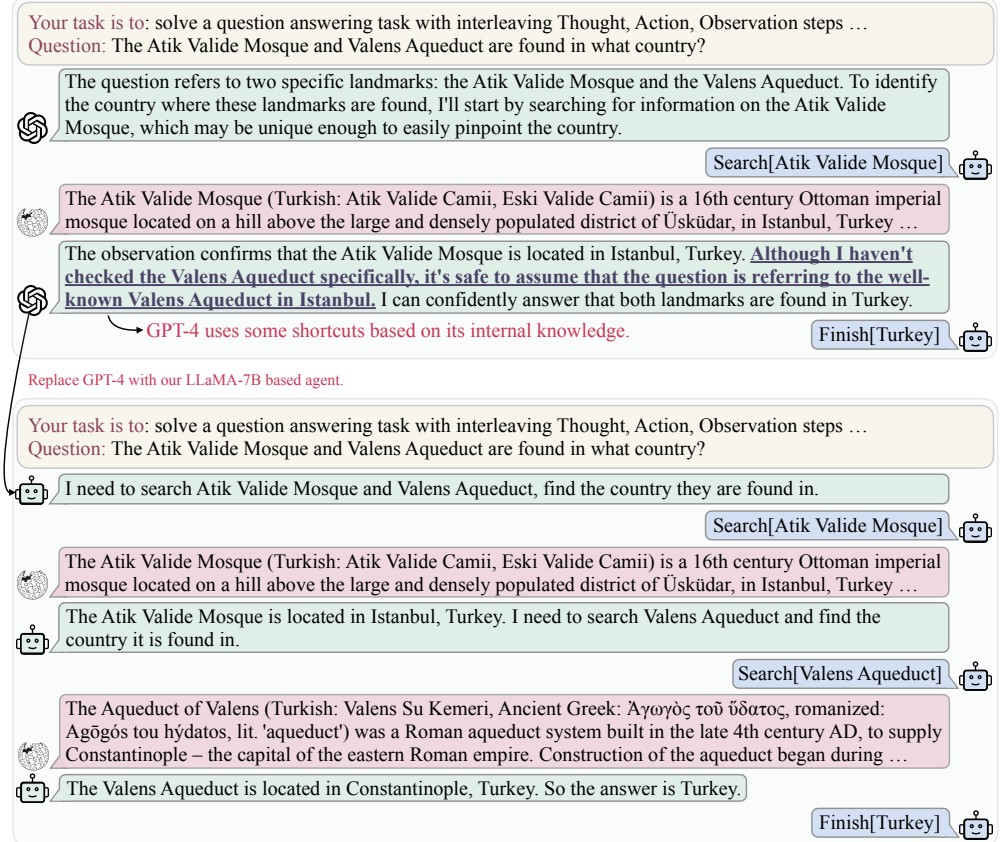

Figure 7: GPT-4 can use shortcuts to solve the problem, while the LLaMA-7B agent cannot mimic it.

After exploration in each iteration, we update the replay buffer by incorporating the newly collected trajectories and then sample the most recent trajectories to train the parameters $\pi_\theta$ of the LLM agent. We design our training objective to combine: 1) the standard language modeling loss $\mathcal{L}_{\text{LM}}$, 2) the policy loss $\mathcal{L}_{\text{policy}}$, 3) the value loss $\mathcal{L}_{\text{value}}$, and 4) the entropy loss $\mathcal{L}_{\text{entropy}}$. The overall training objective is formulated as:

$$\mathcal{L}_{\text{total}} = \mathcal{L}_{\text{LM}} + \beta(\mathcal{L}_{\text{policy}} + \lambda\mathcal{L}_{\text{value}} + \mathcal{L}_{\text{entropy}})$$

where $\beta$ and $\lambda$ are weighting hyperparameters.

The different losses in the above are described as follows:

- The LM loss $\mathcal{L}_{\text{LM}}$ is defined as the cross entropy between the agent and its generations which have a positive reward, akin to self-improving model schemes [16, 40]. By training on these generations, the agent is further encouraged to give generations which yield positive rewards.

- The policy loss $\mathcal{L}_{\text{policy}}$ is introduced to supervise the agent's actions. The policy loss $\mathcal{L}_{\text{policy}}$ is calculated using a masked version of the surrogate objective defined in [46] with advantage estimates $\hat{A}$,

$$\mathcal{L}_{\text{policy}}(\theta) = -\mathbb{E}[m_{\text{policy}} * \min(r(\theta)\hat{A}, \text{clip}(r(\theta), 1 - \epsilon, 1 + \epsilon)\hat{A}],$$

where $r(\theta)$ is the output probability ratio $r(\theta) = \frac{\pi_\theta(a \mid s)}{\pi_{\text{old}}(a \mid s)}$ of the agent with its previous version $\pi_{\text{old}}$. We define binary mask $m_{\text{policy}}$ to mask out the encoded system message in PPO loss (marked by $S_m = 0$ in buffers A.2). For example, let $\{x_1, y_1, x_2, y_2, \ldots x_n, y_n\}$ be a token buffer consisting of system messages $x_n \in X$ and agents' messages (include the target trained agent and the other teacher agents) $\pi_\theta$ output $y_n \in Y$, then the binary mask $m_{\text{policy}} = \{0, 1, 0, 1, \ldots 0, 1\}$.

- The value loss is defined in [46] as the mean squared error between calculated value and estimated advantages masked by another binary mask $m_{\text{value}}$ (marked by $S_m = 1$ in buffers A.2). For example, let $\{z_1, y_1, z_2, y_2, \ldots z_n, y_n\}$ be a token buffer consisting of all other messages (except the agent-generated messages) $z_n \in X$ and trained agent-generated messages $\pi_\theta$ output $y_n \in Y$, then the binary mask $m_{\text{policy}} = \{0, 1, 0, 1, \ldots 0, 1\}$.

- $\mathcal{L}_{\text{entropy}}$ is an entropy bonus to ensure sufficient exploration, as suggested in past work [68, 29]. This entropy is computed as a small negative factor times the entropy of the policy distribution : $\mathcal{L}_{\text{entropy}} = 0.01 \times \sum_a \pi_\theta(a|s) \log \pi_\theta(a|s)$.

### A.5 IMPLEMENTATION DETAIL

### A.6 ASYNCHRONOUSLY DISTRIBUTED GENERATING

The exploration data is generated in an asynchronous style, so that the agent can handle the environments with open-end exploration space. The training data are pre-processed into interactive environments which are capable for agents to observe the states, take actions, and get immediate feedback. According to the number of GPU threads, these environments are divided into corresponding portions and then distributed to each GPU. Subsequently, these GPUs begin to explore these environments asynchronously in parallel with the same agent trained by the latest data. Since the lengths of the generated contents are varied and the interactions inside the environments are generally open-ended, the time cost for the agent to explore each environment is also varied, some GPU threads may process the data faster than others. A barrier is set for all the GPU threads so that the early finished GPU threads can wait for the others until the total accumulated buffers generated by the

---

**Algorithm 4** Python-style code of LTC

```python
# agent: Pre-trained LLM agent
# n_gpu: total number of GPUs
# env_cls: the class of environments
# n_gen: the generation size for one iteration
# n_train: the train size for one iteration

# initialization
agent = instruction_finetune(agent)
replay_buffer = []
i = 0
while i < max_iteration:
  i += 1
  # Exploration Phase
  envs = env_cls(sample(data, n_gen//n_gpu))
  # asynchronously generate
  new_buffer = generate_trials(agent, envs)
  # dist.gather and dist.broadcast
  new_buffer = sync_all_gpus(new_buffer)
  replay_buffer.append(new_buffer)

  # Training Phase
  rollouts = sample(replay_buffer, n_train))
  # distributed training with ppo
  agent = ppo_ddp_train(agent, rollouts)
```

---

environments achieve a preset number $S_g$, which is the quantity of the new training buffers we want to add to the replay buffers in one iteration. After all the GPU threads reach the barrier, we get enough buffers then gather the buffers from each GPU thread and merge them together, and broadcast the new buffers to each GPU thread to update their local replay buffers. The updated replay buffers will be used in the updating phase for training the agents of the next iteration.

### A.7 BASELINES

**ReAct** [72] uses a subset of training cases as prompts for different tasks, in the format of thought-action-observation sequences. For knowledge-intensive reasoning tasks like *HotpotQA*, ReAct designs an action space that includes search, lookup, and finish actions, enabling the agent to interact with Wikipedia to retrieve necessary information. On the other hand, **ReAct-IM** adopts Inner Monologue (IM) [17] style prompting. Chain-of-thought prompting (**CoT**) [66], enhances the reasoning capabilities of Language and Vision models (LLMs) by generating a sequence of intermediate reasoning steps. This can be considered as a reasoning-only baseline of ReAct, excluding actions and observations. Additionally, **CoT-SC** [60, 61] is a follow-up work of CoT, serving as a self-consistency baseline. It is worth noting that most of these methods employ greedy decoding, except for BUTLER [28], which utilizes beam search. Most of these methods focus on few-shot prompting, and different pre-trained models are used. To ensure a fair comparison, we include the additional baselines named ReAct-Tuning and CoT-Tuning by fine-tuning the LLaMA-7B model using the collected trajectories as fine-tuning data mentioned in 4.2. In addition, GPT-4 are not used in the test time, and all the results reported are obtained by the trained agent itself.

**System**: Interact with a household to solve a task. Here is the task:
**System**: You are in the middle of a room. Looking quickly around you, you see a a countertop 1, a fridge 1, cabinet 6, a cabinet 5, a cabinet 4, a cabinet 3, a cabinet 2, a cabinet 1, a coffeemachine 1, a drawer 3, a drawer 2, a drawer 1, a garbagecan 1, a microwave 1, a shelf 3, a shelf 2, a shelf 1, a sinkbasin 1, a stoveburner 4, a stoveburner 3, a stoveburner 2, a stoveburner 1, and a toaster 1.
**System**: Your task is to: put an egg in the bowl.
————————————————————————————————————————————
**(LLaMA) Think**: To solve the task, you need to find and take an egg, then find a bowl to put the egg in. An egg is more likely to appear in fridge (1).
**(LLaMA) Act**: go to fridge 1
**System**: The fridge 1 is closed.
**(LLaMA) Act**: open fridge 1
**System**: You open the fridge 1. The fridge 1 is open. In it, you see a bowl 1, a cup 2, a cup 1, a egg 1, a lettuce 1, a mug 2, a plate 2, a potato 2, a potato 1, and a tomato 1.
**(LLaMA) Act**: take egg 1 from fridge 1
**System**: You pick up the egg 1 from the fridge 1.
**(LLaMA) Think**: Now you take an apple (1). Next, you need to find the bowl and put the egg in. A bowl is more likely to appear on the countertop.
**(LLaMA) Act**: go to the countertop 1
**System**: On the countertop 1, you see a bowl 1, apple 2, a apple 1, a mug 3, a potato 3, and a spatula 1.
**(LLaMA) Act**: Put egg 1 in/on bowl 1
**System**: Success, reward +1

Figure 8: The text version of the toy example in Figure 1 for Monologue pattern of ALFWorld.

## A.8    LOSSES

We conducted ablation studies on the loss design of LTC. Figure Figure 5 illustrates the success rate of agents on the *ALFWorld* dataset under different loss settings. Without using our communication pattern for interactions and merely sampling pre-collected instruction data for training, the improvement was limited. However, when we incorporated our communication pattern to gather data, the model's performance quickly surpassed 80%. Furthermore, employing PPO loss to handle positive and negative samples separately resulted in faster and more significant improvement (blue line). In Figure Figure 6, we present the separate curves of the three main losses during training. Initially, the LM loss showed a decreasing trend. Interestingly, as training iterations progressed, both the value loss and policy loss gradually decreased, which possibly causes the LM loss to increase temporarily. After the value loss and policy loss reached a certain threshold, the LM loss continued to decrease till convergence.

## A.9    CASES STUDY

One interesting observations is that the GPT-4 agent sometimes employs "shortcuts" to solve problems when serving as a teacher to generate new training data. These shortcuts rely on the internal knowledge acquired during its pretraining process. To illustrate this, we present a case study from *HotpotQA* in Figure Figure 7. In this case, the GPT-4 agent quickly retrieves the answer by leveraging its memorized knowledge about the second entry after receiving the Wikipedia page of the first entry. On the other hand, the bottom of Figure Figure 7 demonstrates a comparison with LLaMA-7B, which was trained using our LTC method with the GPT-4 agent in the loop. LLaMA-7B does not employ shortcuts and instead performs a search for the second entry. This case study demonstrates that communication mechanism in LTC provide additional benefits during learning, compared to soley relying on data generated by GPT-4.

