# OpenReview forum: "Adapting LLM Agents Through Communication"
_ICLR.cc/2024/Conference — Submitted to ICLR 2024_

### Official Review · Reviewer_7WiZ · 2023-11-01

**Soundness:** 3 good
**Presentation:** 2 fair
**Contribution:** 3 good
**Rating:** 6
**Confidence:** 3

**Summary:**

The paper proposes a new way of adapting LLMs to new tasks through learning from communication. The authors introduce three communication patterns: Monologue with the LLM interact with the environment and learn from the system provided reward; Dialogue with two LLMs play different roles and the student LLM learn from the teacher's actions; and Analogue with a teacher model provide feedback and reward for student agent's actions. With the different communication patterns, the authors propose to tune the model with both language modeling loss and PPO. Experimental results on three different benchmarks demonstrate the effectiveness of LTC, with additional discussion and ablation validate the design choices.

**Strengths:**

1. The paper introduces a new paradigm of adapting LLMs to downstream tasks that is different from instruction tuning and or prompting. The authors designs three communication patterns considering interaction with the environment and use of multiple LLM agents. While RL with environment and learn from stronger teacher LLMs are things that have been explored in previous works. The paper summarizes these into the three categories, and has additional design to better orchestrate the different components.
2. Experimental results on three different benchmark datasets shows that the proposed learning from communication method can achieve better performance than direct instruction tuning.

**Weaknesses:**

1. The authors proposes three different communication patterns which is nice, however, the experiments only study one pattern for each task. This makes it unclear on what are the pros and cons comparing these three communication patterns, whether each of them could generalize across different tasks and how to choose the right communication pattern. I feel it would be great to have more thorough comparison among the three patterns, and just with the baseline instruction tuning and prompting.
2. The experiment is only done with the 7B model. It is not clear whether the method could apply to smaller models, and more importantly whether it could scale up and how much improvement it could bring to models of larger size.

**Questions:**

1. Have you tried the three patterns across tasks, or they are only tested on the specific task?
2. How many instruction data by GPT 3/4 (as mentioned in section 4.2) are used?

---

> ### Author Response · Authors · 2023-11-23
> **Our Response to Reviewer 7WiZ**
>
> Thank you very much for the constructive comments. We just open-sourced our code in an anonymous Github repo (https://github.com/AIAnonymous/LTC) to help the research community further explore this interesting topic. We'll de-anonymize the repo after the review process.
> 1. Communication patterns across Tasks
>
>     To make the differences among patterns more clear, we reformulate the communication patterns as follows:
>     - (1) **Single-agent Monologue** allows a single agent to generate trajectories in a self-talk style and get interactive feedback from environments;
>     - (2) **Multi-agent Dialogue** enables the multiple agents to interact with each other and external tools, and utilizes the reward signals provided by the environments;
>     - (3) **Teacher-student Dialogue** is a variant of multi-agent dialog in which the teacher agent can provide not only linguistic feedback but also non-linguistic reward signals.
>
>     Different patterns are applied to tasks based on their difficulty and characteristics. In simulated Alfworld environments, the tasks are relatively easy. LLM agents can explore these tasks effectively. For instance, an agent trained with our single-agent Monologue communication achieves a 91% overall success rate. This rate even surpasses GPT-4's success rate of 89.6%. Therefore, there's no need to use multi-agent patterns with GPT-4 as a teacher in this scenario.
>
>     However, as shown in the following table, the situation is different for other tasks. Using only the single-agent Monologue patterns to train the LTC agent shows limited improvements. In these cases, the GPT-4 agent performs much better than our Llama-7B agent. To address this, we designed multi-agent Dialogue and teacher-student Dialogue patterns. These are specifically for HotpotQA and GSM8k tasks. By leveraging GPT-4 agents in the loop, we aim to further improve our agents' training.
>
>     | Method \ Task      |  GSM8K   |  HotpotQA  |  Alfworld |
>     |--------------|--------|------------|-----|
>     | GPT-4 prompting | 87.1   | 55.4  | 89.6 |
>     | Llama-7B-Tuning | 37.7   |  28.2  | 78.4 |
>     | Llama-7B-LTC (Monologue) | 39.6   | 31.0   | 91.0 |
>     | Llama-7B-LTC (Dialogue)    | 41.3 |  33.2  | - |
>
>
> 2. **Scaling up for smaller and larger LLMs**
>
>     To further investigate the scaling-up effectiveness of our method, we conduct extra experiments based on smaller (1B) and larger (13B) models on HotpotQA, the results are shown below:
>
>     | Model           | Method                                             | EM score |
>     |----------------------|----------------------------------------------------|----------|
>     | PaLM-540B       | CoT                                            | 29.4     |
>     |                          | CoT-SC                                         | 33.4     |
>     |                          | ReAct                                          | 27.4     |
>     |                          | ReAct → CoT-SC                                     | 35.1     |
>     | GPT3-175B      | ReAct                                              | 30.8     |
>     | PaLM-62B        | ReAct-Tuning                                       | 32.6     |
>     |                          | CoT-Tuning                                         | 25.2     |
>     | PaLM-8B          | ReAct-Tuning                                       | 25.0     |
>     |                          | CoT-Tuning                                         | 14.0     |
>     | Llama-1B[*](https://github.com/jzhang38/TinyLlama)     | ReAct-Tuning                                       | 12.0     |
>     |                          | LTC                                                | 15.4     |
>     | Llama-7B        | ReAct-Tuning                                       | 28.2     |
>     |                          | LTC                                                | 33.2     |
>     | Llama2-13B   | ReAct-Tuning                                       | 33.8     |
>     |                          | LTC                                                | **35.8** |
>
>
>     It shows that our method with a Llama-2-13B can even surpass the huge PaLM-540B by 0.7%, which is 41x larger. The tiny Llama model with 1B parameters can also benefit from our method, outperforming its tuning baseline. The improvement of our method is consistent when scaling up.
>
>
> 3. **Instruction data**
>
>     For Alfworld datasets, we collect 3.5k ReAct trails on its training set and only use the 1.9k successful trails as the instruction data to fine-tune the agents. For HotpotQA, we use 3.5k successful ReAct trail data from the training set. For GSM8k, we use 7.5k examples with CoT as the instruction fine-tuning data, the same scale as its own training data. We just open-sourced our code in an anonymous Github repo (https://github.com/AIAnonymous/LTC) to help the research community further explore this interesting topic. The instruction finetuning data are also included in this repo for reproducibility.

---

> > ### Comment · Reviewer_7WiZ · 2023-12-05
> >
> > Thanks for the feedback and additional results. After checking comments from other reviewers and author's rebuttal, I would like to maintain my rating as weak accept. Overall the experiments conducted in the paper demonstrates that the proposed method of collecting fine-tuning data and RL is beneficial and consistently outperform regular tuning. I feel the method is worth sharing given the stable gain over baseline. However, there are several places for improvement, either in final version, or revision:
> > - I appreciate the reformulation of the communication pattern, it is more clear now. But still I would recommend  1. improve Figure 1 to make the relationship between patterns more clear. 2. Right now multi-agent & teacher-student are only applied to HotpotQA and GSM8k respectively, as such it is hard to see how these two patterns generalize to different tasks. It would be great to adapt them to both or include more datasets, as also suggest by reviewer 2FJ7.
> > - Model selections, the additional results on smaller/larger models are nice. However, I suggest to make it consistent using either all llama or llama2, as performance of these two versions are significant. Also since GPT4/3 are used, it might be better to include their results as well for reference.

---

### Official Review · Reviewer_SmMH · 2023-11-01

**Soundness:** 2 fair
**Presentation:** 2 fair
**Contribution:** 2 fair
**Rating:** 5
**Confidence:** 4

**Summary:**

This paper proposes a method for adapting LLMs to target domains via RL in domain-specific environments, through distillation from oracle feedback (linguistic and scalar rewards). Evaluation is performed on three NLP tasks (grounded instruction following, multi-hop question answering, and math story reasoning). Results show that the proposed method, which affords exploration during learning with oracle feedback, improves over methods that finetune models only with static domain-specific training data.

**Strengths:**

The paper explores several different tasks and compares against existing methods. It shows relatively strong results showing that exploration and language-like feedback can improve LLMs on domain specific tasks.

**Weaknesses:**

Concern about experimental setup:
* The ablation in Fig 5 doesn't seem to disentangle the contributions of exploration and "communication". In particular, performing exploration and receiving non-linguistic feedback (e.g., in the environment) doesn't seem to be evaluated. And on the other hand, training with static instruction-tuning data augmented with "communication" traces sampled using the same method as LTC (without sampling agent actions, but using the static demonstrations instead) is not evaluated.

Writing feedback: I found the paper was relatively confusing in the terminology used.
* I would suggest renaming the approach away from "through communication". This phrase is very vague, and there really isn't any true communication happening here at all: the learning setting is to just prompt the model(s) to replicate what a dialogue might look like in the domain. It is much more reminiscent of work on distillation and learning from oracles, like DAgger (as briefly mentioned in a footnote on page 7).
* Framing this as continual learning also seems wrong. Fine-tuning with LTC is performed on some held-out set of examples from these datasets' training sets, right? So how is this continual?
* Some of the language around LLMs is too anthropomorphizing. E.g., "human-like" LLM agents, "the agent's brain".
* Minor nitpick, but the whole learning setup is RL, not just the training phase, as the exploration part of the proposed approach is certainly part of a general RL framework.
* There seems to be a bug in Fig 7 with the "question".

One recurring issue with the clarity is that some terminology used is either overly ambiguous, or overly specific. For example:
* "generate trajectories in a self-talk style"
* references to "masks" in the introduction
* "decision-making" as ALFWorld's task -- more precisely, the task is grounded instruction following
* References to a value and a log-prob list in Section 3.1; this is underspecified and seems somewhat irrelevant without having introduced PPO in depth
* "Analogue" as the third communication pattern -- why does it have this name?
* "action tokens" in 4.2
* In the experimental section this is particularly confusing. As a general point, some existing methods are introduced and compared against without intuition on why the experiment is performed (unless the reader is very familiar with the existing approaches). E.g., what data are ReAct-Tuning and CoT-Tuning trained on? There's a self-reference to Section 4.2.

Distinguishing from some existing work would be useful. In particular, work on continual learning for language tasks such as:
* Gao et al. 2022 ("Simulating Bandit Learning from User Feedback for Extractive Question Answering"), and 2023 ("Continually Improving Extractive QA via Human Feedback")
* Kojima et al. 2021 ("Continual Learning for Grounded Instruction Generation by Observing Human Following Behavior")
* More recent work on RLAIF (e.g., Lee et al. 2023, or self-instruct, Yang et al. 2023)

**Questions:**

* The points on the righthand side of Figure 1 seem somewhat arbitrary. ICL is less efficient at inference time, but LTC requires a lot more compute for finetuning.
* How are rewards derived? Is this different in each environment?
* Did you evaluate the quality of the teacher agents? How often are they making mistakes?
* Why not perform a 3x3 experiment combining the three datasets and the three "communication" patterns?
* What are the actual steps for training? It seems the model starts as LLaMA-7B, then is instruction-tuned on a domain-general dataset, then instruction fine-tuned with data sampled from GPT-3/4, and _then_ LTC is applied?
* What is the stopping condition for LTC training? Just when the training data has ran out?
* Why are the results in Table 1 very low precision?

---

> ### Author Response · Authors · 2023-11-23
> **Our Response to Reviewer SmMH (1/2)**
>
> Thank you very much for the constructive comments. We have revised our draft based on your suggestions. We just open-sourced our code in an anonymous Github repo (https://github.com/AIAnonymous/LTC) to help the research community further explore this interesting topic. We'll de-anonymize the repo after the review process.
>
> **Ambiguous terminology**
>
> These comments are very helpful in our revision, now we have updated our draft based on your suggestions. The core change is the definitions of the communication patterns. To make it more clear,  we reformulate the communication patterns as follows:
> - (1) **Single-agent Monologue** allows a single agent to generate trajectories in a self-talk style and get interactive feedback from environments;
> - (2) **Multi-agent Dialogue** enables the multiple agents to interact with each other and external tools, and utilizes the reward signals provided by the environments;
> - (3) **Teacher-student Dialogue** is a variant of multi-agent dialog in which the teacher agent can provide not only linguistic feedback but also non-linguistic reward signals.
>
> **Distinguishing from continual learning**
>
> Yes, we should not frame our methods as continual learning, so we rewrite all parts in the paper that mention continual learning to avoid misleading. With our new formulation of the communication patterns, our current paper focuses on training the LLM agents for single-agent and multi-agent environments.
>
> Experimental setup: performing exploration and receiving non-linguistic feedback
> With our new formulation of the communication patterns, our work mainly targets on studying agent adaptation algorithms in various environments (i.e., single/multi-agent setup). We think the reviewer wants to ask 1. the comparison of “single-agent” v.s. “multi-agent” setup in one particular task. 2. reinforce the learning algorithm with environment reward v.s. (environment reward + feedback signals from other agents).
>
> For “single-agent” v.s. “multi-agent”, we add the experiments on HotpotQA (in Table 2) to show the the comparison between single-agent monologue and multi-agent dialogue, which shows that using Multi-agent Dialogue can further improve the performance:
> | Method      |  EM score  |
> |--------------|--------|
> | Llama-7B-Tuning  |  28.2  |
> | Llama-7B-LTC (Single-agent Monologue) | 31.0   |
> | Llama-7B-LTC (Multi-agent Dialogue)   |  **33.2**  |
>
> For environment reward v.s. (environment reward + feedback signals from other agents), we add the experiments on GSM8k (in Table 3) to show the comparison between single-agent monologue and teacher-student dialogue, which shows that using teacher-student Dialogue can further improve the performance by providing the direct reward signals from the GPT-4 agent:
> | Method      |  Acc  |
> |--------------|--------|
> | Llama-7B-Tuning  |  37.7  |
> | Llama-7B-LTC (Single-agent Monologue) | 39.6   |
> | Llama-7B-LTC (Teacher-student Dialogue)   |  **41.3**  |
>
>
> **Experimental setup: training with static instruction-tuning data**
>
> We want to confirm that you are suggesting two more baselines: 1. SFT on (Instruction-tuning dataset + static few-shot dataset); 2. SFT on (instruction-tuning dataset + communication traces).
> For the 1st baseline, we don’t see the motivation of adding “static demonstration” in the prompt of the training set, since our method does not require few-shot examples as prompt.
> For the 2nd baseline, we present the comparison results in Figure 5, the “w/o PPO” curve can be an example of SFT on communication traces since it uses the exploration traces as the training data and only does SFT on the text part of the data. Our takeaway from the Fig 5 is that what really matters is the communication traces generated by the agent itself, which is better than the SFT data generated by other powerful LLMs.

---

> ### Author Response · Authors · 2023-11-23
> **Our Response to Reviewer SmMH (2/2)**
>
> **Rename the approach**
>
> We appreciate the reviewer's suggestion. In our multi-agent framework, we attribute performance improvement primarily to effective communication, where multiple agents exchange information throughout the process without manual intervention. This is evident in our study, where the student agent, by communicating with teacher agents or the environment, learns more effectively than when limited to static training data.
>
> To further illustrate the role of communication, we introduced the Chameleon game environment in our experiments. Detailed in section 4.2 of our updated paper, the Chameleon game involves two roles: chameleon and non-chameleon. In our setup, all players are played by the Llama2-7B model during training, but in testing, one player is randomly selected to be played by our trained agent, with the others played by GPT-4. The table below showcases the winning rates of our trained agent against GPT-4 players:
>
> | Method \ #players       | n=3    | n=4    | n=5    |  overall |
> |--------------|--------|--------|--------|---------|
>    | Tuning | 20.8   | 20.3   | 23.8   | 21.9    |
> | LTC (multi-agent dialogue)   | **22.9** | **23.4** | **27.5** | **25.0**  |
>
> The results show that our LTC method surpasses instruction tuning baselines by 3.1% in overall winning rates. This underscores the importance of communication in the Chameleon game, which demands bluffing, deduction, and strategic interaction between agents and their environment for success.
>
>
> **The training phase**
>
> Sure, we’ve renamed the Training Phase as the Updating Phase in the text and figures. Now the iterative training is a standard reinforcement learning pipeline, which has the exploration phase and the updating phase.
>
>
> **The bug in Figure 7**
>
> Thanks for reminding us, the question text is forgotten to replace when using the template of Fig3, we’ve updated it with the correct question text.
>
>
> **The righthand side of Figure 1**
>
> Yes, the original Figure 1 does not properly show our core ideas. So we’ve removed the original Figure 1 and changed it to the current Figure 1 with a more clear diagram to show our LTC could process both single-agent and multi-agent environments, which is the core of our method.
>
>
> **Experiments combining the datasets and patterns**
>
> Different patterns are applied to tasks based on their difficulty and characteristics. In simulated Alfworld environments, the tasks are relatively easy. LLM agents can explore these tasks effectively. For instance, an agent trained with our single-agent Monologue communication achieves a 91% overall success rate. This rate even surpasses GPT-4's success rate of 89.6%. Therefore, there's no need to use multi-agent patterns with GPT-4 as a teacher in this scenario.
> However, as shown in the following table, the situation is different for other tasks. Using only the single-agent Monologue patterns to train the LTC agent shows limited improvements. In these cases, the GPT-4 agent performs much better than our Llama-7B agent. To address this, we designed multi-agent Dialogue and teacher-student Dialogue patterns. These are specifically for HotpotQA and GSM8k tasks. By leveraging GPT-4 agents in the loop, we aim to further improve our agents' training.
>
> | Method \ Task      |  GSM8K   |  HotpotQA  |  Alfworld |
> |--------------|--------|------------|-----|
> | GPT-4 prompting | 87.1   | 55.4  | 89.6 |
> | Llama-7B-Tuning | 37.7   |  28.2  | 78.4 |
> | Llama-7B-LTC (Monologue) | 39.6   | 31.0   | 91.0 |
> | Llama-7B-LTC (Dialogue)    | 41.3 |  33.2  | - |
>
> **Reward**
>
> For the single-agent monologue and multi-agent dialogue patterns, the rewards are provided by the environments. For the teacher-student dialogue pattern, the rewards come from both the environment and the teacher agent.
>
> **The actual steps for training**
>
> There are only two steps starting from the pre-trained LLMs(I.e, Llama):
> - 1. We use the collected task-specific instruction fine-tuning data to fine-tune the Llama to help it learn the instruction-following ability (like ReAct/CoT). And the output models are our Llama-Tuning baseline in our experiments.
> - 2. We put the fine-tuned agents in our communication patterns to explore the environments and collected the traces for training iteratively.
>
> **Stopping condition**
>
> In our experiments, we set the stopping condition as a max iteration number, and this number is chosen considering our computation resource and the performance of the model. This number could theoretically be increased to infinity in the case of online learning, though it is unclear if that would have unintended side effects.
>
> **low-precision in Table 1**
>
> The top half of the Table is obtained from Table 3 of the ReAct Paper, which uses the same precision, so we follow their settings for a fair comparison.

---

### Official Review · Reviewer_2FJ7 · 2023-11-02

**Soundness:** 3 good
**Presentation:** 3 good
**Contribution:** 3 good
**Rating:** 6
**Confidence:** 4

**Summary:**

The paper presents a new training paradigm called Learning through Communication (LTC), which enables large language model (LLM) agents to adapt to new tasks through interaction. The major contributions include the LTC method itself, the introduction of task-specific communication patterns (Monologue, Dialogue, Analogue), and empirical evidence that LTC outperforms instruction-tuning baselines on decision-making, knowledge-intensive reasoning, and numerical reasoning tasks. The paper demonstrates LTC's effectiveness and efficiency, with significant gains in success rates and accuracy, along with reduced token usage during inference.

**Strengths:**

Novelty of the Learning Method: The proposed Learning through Communication (LTC) framework is a commendable advancement. It is an inventive approach that enables language models to dynamically adapt to new tasks through iterative interactions. The methodology is well-conceived, blending language modeling with reinforcement learning objectives in a manner that is both theoretically sound and practically viable.

Comprehensive Evaluation: The authors have conducted a rigorous empirical evaluation of the LTC framework across a variety of tasks and datasets. The breadth of the evaluation—spanning decision-making, knowledge-intensive reasoning, and numerical reasoning—is impressive. This comprehensive testing not only demonstrates the applicability of LTC to a wide range of tasks but also provides a convincing argument for its efficacy compared to existing baselines.

**Weaknesses:**

Model Comparisons: The LTC method's performance is compared to that of models with and without tuning. However, the paper states that the combined method of ReAct and CoT-SC surpasses LTC by 1.9%. This suggests that while LTC has strengths, there may be specific configurations of existing methods that outperform it, which could be a point of concern regarding the robustness and superiority of LTC​1​.

Scope of Evaluation: While LTC is shown to perform well across three tasks, the evaluation might still be limited in scope. The paper hints at future work to explore more diverse communication patterns and involve communication with humans. This suggests that the current evaluation may not fully capture the LTC's performance in more varied or complex interactive settings​ such as ScienceWorld, and Mind2Web.

Generalization to Human Interaction: The paper outlines future work to involve communication with humans during the learning process, which is not covered in the current evaluation. This omission indicates that the paper does not address the challenge of human-agent interaction, which is critical for practical applications of LLM agents

Minor point:
References that can be helpful for discussion:
1) ScienceWorld: Is your Agent Smarter than a 5th Grader?
2) Mind2Web: Towards a Generalist Agent for the Web
3) AgentBench: Evaluating LLMs as Agents
4) SwiftSage: A Generative Agent with Fast and Slow Thinking for Complex Interactive Tasks
5) ReWOO: Decoupling Reasoning from Observations for Efficient Augmented Language Models

**Questions:**

1. The LTC method relies heavily on predefined communication patterns. Have the authors considered how LTC might generalize to tasks that require more flexible or less structured forms of communication?

2. Given that certain combined methods like ReAct and CoT-SC have outperformed LTC, what are the authors' perspectives on the limitations of LTC in its current form? Are there specific enhancements they are considering to improve upon these existing methods?

3. The paper suggests future work will involve human communication. Can the authors provide preliminary insights into how they expect human-in-the-loop interactions to affect the learning process and the adaptability of the LTC method?

---

> ### Author Response · Authors · 2023-11-22
> **Our Response to Reviewer 2FJ7 (1/2)**
>
> Thank you very much for the constructive comments. We just open-sourced our code in an anonymous Github repo (https://github.com/AIAnonymous/LTC) to help the research community further explore this interesting topic. We'll de-anonymize the repo after the review process.
>
> 1. **Model Comparisons**
>
>     We thank the reviewer for noting the comparison between LTC and other methods such as ReAct and CoT-SC in Table 2. We would like to point out that results on ReAct and CoT-SC are reported based on the PaLM (540B) model, while LTC is on Llama (7B). To further investigate the scaling-up effectiveness of our method, we conduct extra experiments based on smaller (1B) and larger (13B) models on HotpotQA, the results are shown below:
>
>     | Model           | Method                                             | EM score |
>     |----------------------|----------------------------------------------------|----------|
>     | PaLM-540B       | CoT                                            | 29.4     |
>     |                          | CoT-SC                                         | 33.4     |
>     |                          | ReAct                                          | 27.4     |
>     |                          | ReAct → CoT-SC                                     | 35.1     |
>     | GPT3-175B      | ReAct                                              | 30.8     |
>     | PaLM-62B        | ReAct-Tuning                                       | 32.6     |
>     |                          | CoT-Tuning                                         | 25.2     |
>     | PaLM-8B          | ReAct-Tuning                                       | 25.0     |
>     |                          | CoT-Tuning                                         | 14.0     |
>     | Llama-1B[*](https://github.com/jzhang38/TinyLlama)     | ReAct-Tuning                                       | 12.0     |
>     |                          | LTC                                                | 15.4     |
>     | Llama-7B        | ReAct-Tuning                                       | 28.2     |
>     |                          | LTC                                                | 33.2     |
>     | Llama2-13B   | ReAct-Tuning                                       | 33.8     |
>     |                          | LTC                                                | **35.8** |
>
>
>     It shows that our method with a Llama-2-13B can even surpass the huge PaLM-540B by 0.7%, which is 41x larger. The tiny Llama model with 1B parameters can also benefit from our method, outperforming its tuning baseline. The improvement of our method is consistent when scaling up.
>
>
> 2. **Scope of Evaluation**
>
>     We conducted some extra experiments on a more challenging multi-agent environment setup (i.e., Chameleon Game). Preliminary results are updated in the paper ( pls see section 4.2 for details).
>     There are two roles in the [Chameleon](https://github.com/Farama-Foundation/chatarena) game: chameleon and non-chameleon. The topic of the secret word will be first revealed to all the players. Then the secret word will be revealed to non-chameleons. Non-chameleons try to identify the chameleon without giving away the secret word, while the chameleon tries to blend in and guess the word.
>     In our training, all the players are played by the same Llama2-7B model. While in the testing, to get the winning rate of our trained agent against GPT4, only 1 player is randomly picked to use our trained agent as the backend, and other players are played by GPT4. The following results are their winning rate against GPT-4 players:
>
>     | Method \ #players   | n=3    | n=4    | n=5    |  overall |
>     |-----------|--------|--------|--------|---------|
>     | Tuning | 20.8   | 20.3   | 23.8   | 21.9    |
>     | LTC    | **22.9** | **23.4** | **27.5** | **25.0**  |
>
>
>     As shown above, LTC outperforms the instruction tuning baselines by 3.1% overall winning rates. To further clarify, we aren’t trying to demonstrate that multi-agent setup is superior to single-agent setup in experiments, but we show that “learning through communication” allows agents to be online-trainable in various complex single/multiple agent environments through its conversation data and feedback signals.

---

> ### Author Response · Authors · 2023-11-22
> **Our Response to Reviewer 2FJ7 (2/2)**
>
> 3. **flexible forms of communication**
>
>     Great question. We explored recent open source multi-agent libraries (i.e., AutoGen, ChatArena, Ai-Town etc), however most of them require users to predefine the communication patterns as well, while some of them define some randomness in the environment to allow the agent to communicate with another random agent (i.e., AI-town). We are actively exploring to build a large-scale multi-agents environment for simulating general collaboration and competition tasks between random-agents. We are hoping such a sandbox system could be a good testbed for comparing different LTC strategies.
>
> 4. **Limitation of LTC**
>
>     As shown above, we have demonstrated that by scaling up LTC to Llama2-13B, we can achieve superior results, surpassing even the larger PaLM-540B combined methods. This underscores the scalability and potential of LTC. Actually our LTC is orthogonal to these promising methods like ReAct and CoT, which focus on prompting and we are focusing on training. So the integration of our methods with these advanced prompting methods to improve the communication effectiveness is what we are investigating to further enhance our framework.
>     Of course there are also some limitations of LTC:
>     - Cost: computation resource requirement would be huge if we allow personal assistant agents to be trainable (alternative solution could be hosting LTC agents on personal edge device)
>     - Stability: LTC adopts an RL approach for iterative learning. Thus it doesn’t guarantee to procedure a better-performance model after every epoch of training.
>
> 5. **Generalization to Human Interaction**
>
>     The integration of human-in-the-loop interactions with the LTC agent could significantly enhance the model's adaptability and learning process, particularly in terms of customization and personalization. By directly incorporating human feedback, LTC can tailor its responses more effectively to individual user preferences and requirements. This approach not only ensures that the technology adapts to the specific needs and contexts of different users but also improves user engagement and satisfaction. Such personalized interaction is crucial for the broad applicability of LTC in diverse practical scenarios, where unique user experiences and customized responses are highly valued.
>
>
> 6. **References**
>
>     Thanks for providing these helpful References, we’ve add them into our discussion of related works

---

> > ### Comment · Reviewer_2FJ7 · 2023-12-04
> >
> > Thank you for the feedback. After reviewing both the other reviews and the authors' rebuttal, I have assessed the overall quality of the paper, and I intend to maintain my original score.

---

### Official Review · Reviewer_CuLf · 2023-11-09

**Soundness:** 3 good
**Presentation:** 3 good
**Contribution:** 2 fair
**Rating:** 5
**Confidence:** 4

**Summary:**

The paper proposes a method that uses LLM based agents to solve tasks that require reasoning and sequential decision making. Authors propose a paradigm for learning through communication where communication can be 1.) monologue - self communication, 2.) dialogue - multi-agent communication or 3.) analogue - communicating with a teacher agent. Authors provide results on three environments  1.) ALFworld - sequential decision making environment that require performing household tasks, 2.) GSM8K - grade school math problem solving and 3.) HotPotQA - reasoning and language understanding.

**Strengths:**

- The paper is well written and easy to read.
- The paper provides a  comprehensive summary of related work and the contributions of the paper are well placed in the relevant literature
- Environments considered in the paper are diverse and capture essential features of reasoning and sequential decision making

**Weaknesses:**

- Multi-agent interactions highlighted in the paper are not properly formulated and motivated.
- Authors are not considering any environments that require multi-agent interactions. This reviewer is not convinced that the paper adds value in terms of communication between multiple agents that would lead to better performance.
- Authors have not provided the code and hence the results are not reproducible.

**Questions:**

- According to the provided discussion the only difference between dialogue and analogue is in analogue, teacher roles can directly provide reward signals and new examples. What is the significance of this distinction?
- How does this approach extend to multi-agent environments?

---

> ### Author Response · Authors · 2023-11-22
> **Our Response to Reviewer CuLf**
>
> Thank you very much for the constructive comments. We just open-sourced our code in an anonymous Github repo (https://github.com/AIAnonymous/LTC) to help the research community further explore this interesting topic. We'll de-anonymize the repo after the review process.
>
>  1. ____The multi-agent environments____
>
>     Thanks for your suggestion. We conducted some extra experiments on a multi-agent environment setup (i.e., Chameleon Game). Preliminary results are updated in the paper ( pls see section 4.2 for details). There are two roles in the [Chameleon](https://github.com/Farama-Foundation/chatarena) game: chameleon and non-chameleon. The topic of the secret word will be first revealed to all the players. Then the secret word will be revealed to non-chameleons. Non-chameleons try to identify the chameleon without giving away the secret word, while the chameleon tries to blend in and guess the word.
>     In our training, all the players are played by the same Llama2-7B model. While in the testing, to get the winning rate of our trained agent against GPT4, only 1 player is randomly picked to use our trained agent as the backend, and other players are played by GPT4. The following results are their winning rate against GPT-4 players:
>
>     | Method \ #players       | n=3    | n=4    | n=5    |  overall |
>     |--------------|--------|--------|--------|---------|
>     | Tuning | 20.8   | 20.3   | 23.8   | 21.9    |
>     | LTC    | **22.9** | **23.4** | **27.5** | **25.0**  |
>
>
>     As shown above, LTC outperforms the instruction tuning baselines by 3.1% overall winning rates. To further clarify, we aren’t trying to demonstrate that multi-agent setup is superior to single-agent setup in experiments, but we show that “learning through communication” allows agents to be online-trainable in various complex single/multiple agent environments through its conversation data and feedback signals.
>
>
> 2. _____Multi-agent interactions are not properly formulated_____
>
>     To further highlight the multi-agent interaction and make it more clear, we reformulate the communication patterns as follows:
>     - (1) **Single-agent Monologue** allows a single agent to generate trajectories in a self-talk style and get interactive feedback from environments;
>     - (2) **Multi-agent Dialogue** enables the multiple agents to interact with each other and external tools, and utilizes the reward signals provided by the environments;
>     - (3) **Teacher-student Dialogue** is a variant of multi-agent dialog in which the teacher agent can provide not only linguistic feedback but also non-linguistic reward signals.
>
> 3. ____The difference between Dialogue and Analogue____
>
>     As shown in our new multi-agent formulations, both the second and third patterns are designed for dialogue between multiple agents. The difference is that the agents in the second pattern can only receive the rewards provided by the environments, while the Teacher-student Dialogue patterns enable the teacher agent to directly give the reward to the student.

---

### Author Response · Authors · 2023-11-22
**General Response: Revisionn Uploaded and Code Released**

We thank all reviewers for their comments. In addition to the specific response below, here we summarize our goal and the changes planned to be included in the revision. Specific changes include:


1. **We release our code**

We just open-sourced our code in an anonymous Github repo (https://github.com/AIAnonymous/LTC) to help the research community further explore this interesting topic. We'll de-anonymize the repo after the review process.



2. **Our framework is also effective for smaller and larger LLMs**

To further investigate the scaling-up effectiveness of our method, we conduct extra experiments based on smaller (1B) and larger (13B) models on HotpotQA, the results are shown below:

 | Model           | Method                                             | EM score |
|----------------------|----------------------------------------------------|----------|
| PaLM-540B       | CoT                                            | 29.4     |
|                          | CoT-SC                                         | 33.4     |
|                          | ReAct                                          | 27.4     |
|                          | ReAct → CoT-SC                                     | 35.1     |
| GPT3-175B      | ReAct                                              | 30.8     |
| PaLM-62B        | ReAct-Tuning                                       | 32.6     |
|                          | CoT-Tuning                                         | 25.2     |
| PaLM-8B          | ReAct-Tuning                                       | 25.0     |
|                          | CoT-Tuning                                         | 14.0     |
| Llama-1B[*](https://github.com/jzhang38/TinyLlama)     | ReAct-Tuning                                       | 12.0     |
|                          | LTC                                                | 15.4     |
| Llama-7B        | ReAct-Tuning                                       | 28.2     |
|                          | LTC                                                | 33.2     |
| Llama2-13B   | ReAct-Tuning                                       | 33.8     |
|                          | LTC                                                | **35.8** |

It shows that our method with a Llama-2-13B can even surpass the huge PaLM-540B by 0.7%, which is 41x larger. The tiny Llama model with 1B parameters can also benefit from our method, outperforming its tuning baseline. The improvement of our method is consistent when scaling up.



3. **Our framework is also applicable to multi-agent conversation environments**

We conducted some extra experiments on a multi-agent environment setup (i.e., Chameleon Game). Preliminary results are updated in the paper ( pls see section 4.2 for details).
There are two roles in the [Chameleon](https://github.com/Farama-Foundation/chatarena) game: chameleon and non-chameleon. The topic of the secret word will be first revealed to all the players. Then the secret word will be revealed to non-chameleons. Non-chameleons try to identify the chameleon without giving away the secret word, while the chameleon tries to blend in and guess the word.
In our training, all the players are played by the same Llama2-7B model. While in the testing, to get the winning rate of our trained agent against GPT4, only 1 player is randomly picked to use our trained agent as the backend, and other players are played by GPT4. The following results are their winning rate against GPT-4 players

| Method \ #players       | n=3    | n=4    | n=5    |  overall |
|--------------|--------|--------|--------|---------|
   | Tuning | 20.8   | 20.3   | 23.8   | 21.9    |
| LTC    | **22.9** | **23.4** | **27.5** | **25.0**  |


As shown above, LTC outperforms the instruction tuning baselines by 3.1% overall winning rates. To further clarify, we aren’t trying to demonstrate that multi-agent setup is superior to single-agent setup in experiments, but we show that “learning through communication” allows agents to be online-trainable in various complex single/multiple agent environments through its conversation data and feedback signals.







4. **We reformulate the communication patterns**

The LTC method is a general learning paradigm that can be applied to diverse agent settings with single-agent and multi-agent environments. To collect communication trajectories and reward signals, we designed three structured communication patterns:
- (1) **Single-agent Monologue** allows a single agent to generate trajectories in a self-talk style and get interactive feedback from environments;
- (2) **Multi-agent Dialogue** enables the multiple agents to interact with each other and external tools, and utilizes the reward signals provided by the environments;
- (3) **Teacher-student Dialogue** is a variant of multi-agent dialog in which the teacher agent can provide not only linguistic feedback but also non-linguistic reward signals.

---

### Meta-Review · Area_Chair_UXfi · 2023-12-06

**Metareview:**

This paper introduces LTC, a method to adapt LLM-based agents to solve reasoning/tool-use tasks through various types of “communication” patterns: single-agent monologue, similar to chain-of-thought reasoning, multi-agent “dialogues” where a set of agents exchange information to collaboratively solve a task or compete with each other, as well as “teacher-student” dialogues where a student agents receives textual feedback from a teacher model. The authors proposed a learning algorithm that first performs exploration by sampling communication trajectories via the interaction between agents and environments, and then update the model using the sampled trajectories and their reward using a PPO algorithm. Results on four environments demonstrate the effectiveness of LTC.

**Strengths**

This paper attempts to unify several agent-based tasks as learning via communication between agents or environments, with relatively strong empirical results.

**Open Weaknesses**
* Some key technical terms used in this paper are vague and confusing (SmMH). First of all, the terms **communication** and **dialogue** used in the paper may suggest that this paper focuses on scenarios where agents interact with each other via natural language utterances (besides the monologue pattern). However, not all of the tasks follow this pattern, with the HotpotQA task being an exception, where only the “thinker” provides textual instructions to another agent, while the response from the other agent (blue boxes in Figure 3, Left) is not in natural language. Therefore, it would be inappropriate to name interactions in HotpotQA as “multi-agent dialogues”. While the authors attempted to fix issues with using vague terminologies by defining a taxonomy of agent communication patterns per feedback from SmMH, this issue still remains in the current version, and it would require significant revisions to fix this (e.g., by renaming multi-agent/teacher-student “dialogues” with other terms, or removing the experiments on HotpotQA). Besides, there are other minor issues w.r.t. the representation that could be mis-leading, such as the last sentence in the fourth paragraph, Sec 1 (“Teacher-student dialogue ... *not only* linguistic feedback … ”). In fact, the “Teacher-student dialogue” on GSM8K is the *only* task where agents will receive linguistic feedback (Figure 3, Right, green boxes).

* Missing important ablations: to demonstrate the additional value of **communication**, the authors should compare LTC with other RL/self-training/exploration approaches that sample target trajectories as training examples based on environment feedback but without involving any communications (e.g., monologues). For example, predicting actions in the blue boxes in Figure 3 (Left) without the thinker’s CoT, or predicting only the textual navigation actions in ALFWorld without monologues. This issue was also related to SmMH’s comment on the “concern about experimental setup”.

* Only one environment is considered for each four communication patterns (7WiZ), making it hard to assess if LTC is a general approach for different tasks from each communication pattern.

* This paper does not consider more complex communication paradigms, such as human-bot dialogue (2FJ7)

**Resolved Issues**

* The multi-agent interactions highlighted in the paper are not properly formulated, with no experiments (CuLf). This seems a significant issue with the initial version, and was partially resolved after including the Chameleon experiments.

* Evaluation on larger Llama models (7WiZ) — resolved after including Llama-13B experiments.

**Justification For Why Not Higher Score:**

Given that there are still major issues regarding the technical presentation and ablations, the recommendation is a "Reject".

**Justification For Why Not Lower Score:**

N/A

---

### Decision · Program_Chairs · 2024-01-16

Reject